# Developing a Biodegradable Film for Packaging with Lignocellulosic Materials from the Amazonian Biodiversity

**DOI:** 10.3390/polym15173646

**Published:** 2023-09-04

**Authors:** Danillo Wisky Silva, Felipe Gomes Batista, Mário Vanoli Scatolino, Adriano Reis Prazeres Mascarenhas, Dayane Targino De Medeiros, Gustavo Henrique Denzin Tonoli, Daniel Alberto Álvarez Lazo, Francisco de Tarso Ribeiro Caselli, Tiago Marcolino de Souza, Francisco Tarcísio Alves Junior

**Affiliations:** 1Department of Production Engineering, State University of Amapá (UEAP), Post-Graduate Program in Intellectual Property and Technology Transfer for Innovation (PROFNIT), Macapá 68900-070, AP, Brazil; danilowisky@hotmail.com (D.W.S.); tarcisioalvesjr@yahoo.com.br (F.T.A.J.); 2Department of Forest Sciences, Federal University of Lavras—UFLA, Lavras 37203-202, MG, Brazil; felipejp.gomes@gmail.com (F.G.B.); dayanemedeirost@gmail.com (D.T.D.M.); gustavotonoli@ufla.br (G.H.D.T.); 3Department of Agronomic and Forest Sciences, Federal Rural University of the Semi-Arid (UFERSA), Mossoró 59625-900, RN, Brazil; 4Department of Forest Engineering, Federal University of Rondônia (UNIR), Rolim de Moura 76801-059, RO, Brazil; adriano.mascarenhas@unir.br; 5Forest Department, University of Pinar del Río “Hermanos Saiz Montes de Oca”, Pinar del Río 20100, Cuba; daniel@upr.edu.cu; 6Department of Forest Sciences, Federal University of Piauí (UFPI), Post-Graduate Program in Intellectual Property and Technology Transfer for Innovation (PROFNIT), Teresina 64049-550, PI, Brazil; tarso.caselli@ufpi.edu.br; 7Department of Production Engineering, State University of Amapá (UEAP), Macapá 68900-070, AP, Brazil; tiago.souza@ueap.edu.br

**Keywords:** Amazonian fibers, barrier properties, ecological packaging, mechanical fibrillation

## Abstract

The development of packaging films made from renewable raw materials, which cause low environmental impact, has gained attention due to their attractive properties, which have become an exciting option for synthetic films. In this study, cellulose micro/nanofibrils (MFC/NFC) films were produced with forest residues from the Amazon region and evaluated for their potential to generate alternative packaging to traditional plastic packaging. The MFC/NFC were obtained by mechanical fibrillation from fibers of açaí seeds (*Euterpe oleracea*), titica vine (*Heteropsis flexuosa*), and commercial pulps of *Eucalyptus* sp. for comparison. The fibrillation of the titica vine culminated in higher energy expenditure on raw materials. The açaí films showed a higher tensile strength (97.2 MPa) compared to the titica films (46.2 MPa), which also showed a higher permeability rate (637.3 g day^−1^ m^−2^). Films of all raw materials scored the highest in the grease resistance test (n° 12). The films produced in the study showed potential for use in packaging for light and low moisture products due to their adequate physical, mechanical, and barrier characteristics. New types of pre-treatments or fibrillation methods ecologically correct and viable for reducing energy consumption must be developed, mainly for a greater success of titica vine fibrillation at the nanoscale.

## 1. Introduction

The diversity of vegetal raw materials in the Brazilian Amazon region can still be better exploited to obtain new wood and non-wood products. Among the raw materials, cellulose stands out, as it is the most abundant polymer found in nature, inexpensive, and non-toxic. Due to its attractive properties, it has increasingly attracted the interest of the scientific community and the industrial sector [1]. Among the products that can be obtained from cellulose, are the micro/nanofibrils (MFC/NFC), which can be extracted from different lignocellulosic sources using mechanical, chemical, and enzymatic methods, or even by a combination of methods [2]. Mechanical processes generally result in higher yields and less degradation of cellulose during extraction, while chemical methods may produce lower yields. The MFC/NFC have a greater surface area, facilitating the connection between the fibrils. Therefore, they can generate unique and more resistant papers, special packaging, coatings, and composites for the automotive, aerospace, and sports industries.

Some processes are required for successful MFC/NFC extraction and better product quality at the nanoscale. Alkaline treatment and bleaching have become increasingly common options for removing non-cellulosic components from natural fibers [3], as they use common and low-cost reagents. If the pre-treatments are unsatisfactory, the fibrous material can be morphologically damaged during the fibrillation stages.

When biomass is incubated in alkali reagents, the cellulose swells, breaking its crystalline structure, generating a greater porosity, and exposing a larger surface area. Additionally, ester linkages and, to a lesser degree, ether linkages between hemicellulose and lignin are broken under alkaline conditions, which significantly increases their solubility and leads to a decrease in the degree of polymerization [4].

The intrinsic characteristics of each type of fiber require adjustments in the pre-treatment conditions, such as solvent level, reagent concentration, temperature, and reaction time. In inadequate parameters, the resulting MFC/NFC may be low-quality [5]. The primary source for MFC/NFC production has been commercial kraft pulp, mainly from *Eucalyptus* sp. wood, as it is the main product of planted forests for pulp production purposes in Brazil.

The uses of kraft pulp range from paper for packaging and personal hygiene (toiletries, diapers, pads, paper towels, and napkins) to writing and printing papers [6]. However, the use of other plant fibers has also been explored, such as bamboo fibers [7], oat straw [8], açaí fibers [9], pseudostem banana [6], sugar beet pulp chips [10], sunflower stem pith [11], and mandacaru cactus [1].

Because of the number of lignocellulosic fiber sources offered due to the biodiversity in the Amazon, it can be considered that many species still need to have their potential evaluated in the field of technology, especially considering the nanoscale. Açaí (*Euterpe oleracea* Mart.), belonging to the Arecaceae family, has several economic prospects, mainly food with the consumption of its purplish-colored pulp [12]. The fibers are a byproduct of pulping and are present in the lining of the mesocarp (seed) and do not have a particular destination. They are used only for decoration and handicrafts for regional producers. These lignocellulosic materials are still treated as waste in the production chain and end up being incorrectly dumped into the environment, becoming land and water pollutants when, for example, they are discharged to the banks of springs, causing a reduction in dissolved oxygen in the water and eutrophication [13].

Another species from the Amazon that could still be studied in the field of nanotechnology is the popularly called titica (*Heteropsis flexuosa*), belonging to the Araceae family. Its natural occurrence seems to indicate a healthy forest because more light penetration through the forest canopy to the soil inhibits the occurrence of the species [14]. The fibers are extracted from aerial roots thrown to the ground, and when they reach the surface, they are thick, woody, resistant, and durable. *Heteropsis* roots are distinguished from other vigorous vines. They are particularly attractive for commercial purposes because their dark epidermis and cortex can be easily peeled away to leave a rot-resistant structure packed with long, strong fibers.

This core can be further split along the vine’s length into slender pieces that retain strength and flexibility [15]. They were included in the market as excellent raw materials due to their economic importance for manufacturing various products, such as handicrafts, baskets, and furniture [16]. Knowledge about the species for application in technology is still scarce and a great challenge for Brazilian researchers, who must thoroughly know the quality of regional species.

Petroleum-derived plastics are cheap and easily obtainable, but the plastic degradation time in the environment is very long, indicating a problem in marine ecosystems. Thus, this work brings an initiative on applying species representing Amazonian biodiversity in packaging technology with a low pollutant arsenal and potential for replacing petroleum- and mineral-based products. This research aimed to produce and characterize MFC/NFC suspensions and films with added value from residual fibers derived from extracting açaí seeds and titica vine roots of renewable origin. Additionally, the study emphasized aspects of pre-treatments and the need, or not, to carry them out for good quality MFC/NFC films for applications in quality packaging for light products, such as slices of bread, biscuits, and light and moisture-free flours.

## 2. Material and Methods

### 2.1. Material

The açaí fibers were manually removed from seeds collected in a commercial establishment in Paragominas, Pará, Brazil. The seeds were from fruits obtained from açaí plantations, commonly used for extracting the pulp. Titica vine roots were obtained from local commerce in Macapá, Amapá, Brazil. The vine-shaped stalks were cut to a length of ~15 cm and immersed in water for 24 h to hydrate and swell the fiber and, consequently, facilitate fibrillation. The bleached cellulose pulps of Eucalyptus sp. were obtained by the kraft chemical pulping process supplied by Suzano Papel e Celulose (Suzano—SP, Brazil).

### 2.2. Fiber Treatment

The alkaline treatment of açai fibers, titica, and commercial pulp of *Eucalyptus* sp. and their bleaching was carried out according to the methodology suggested by Aguado et al. [17]. For every 5 g of fibers of each material, 100 mL of 5% NaOH solution (*w*/*w*) were used. The fibers and pulps were treated in a water bath at 80 °C under agitation at 650 rpm for 2 h. Subsequently, the samples were washed with deionized water until pH~7. In addition to the alkaline pre-treatment, the açaí fibers underwent bleaching, being placed in 100 mL of a 1:1 (*v*/*v*) solution of 24% (*v*/*v*) H_2_O_2_ and 4% (*w*/*v*) NaOH for 2 h at 80 °C in a water bath, with mechanical agitation at 650 rpm. After filtering to remove excess water with the reagents, the fibers were washed with deionized water and a 20% (*v*/*v*) acetic acid solution until reaching pH~7.

### 2.3. Chemical Composition of Fibers

The fibers were characterized in terms of extractive contents: Tappi T 204 om-97 [18]; lignin: Tappi T 222 om-02 [19]; holocellulose (cellulose + hemicelluloses): Browning [20]; cellulose: Kennedy et al. [21]; hemicelluloses: (holocellulose − cellulose); and ashes: Tappi T 211 om-02 [22]. Average values for each component analysis were obtained from 3 replicates.

### 2.4. Production of Cellulose Micro/Nanofibrils (MFC/NFC)

For the production of MFC/NFC, each material was immersed in deionized water at a concentration of 1.5% (m/m) for fiber hydration and pulp in the case of commercial eucalyptus. The raw materials were subjected to fibrillation using the SuperMasscolloider Masuko Sangyo MKCA6-3 ultra-refiner (Honcho, Japan). The MFC/NFC were obtained after reaching a gelatinous appearance, using an opening between the discs ranging from 10 to 100 µm and a rotation of 1500 rpm, following procedures reported in previous works [23]. Fibrillation is a process that transforms fibers from lignocellulosic material from the microscale to the nanoscale. The fibrillator mill has 2 stone discs, one rotating and one static, that promote the disruption of the plant cell wall by shear forces. The fibrillated state means that the nanofibrils can make stronger connections.

### 2.5. Energy Consumption

In mechanical fibrillation, the energy consumption of the processing was obtained until each material reached a gelatinous appearance when considering the average electric current (A) measured during the grinder operation. Energy consumption (kWh/t) was calculated by Equation (1), according to the study by Scatolino et al. [9].
(1)$$\mathrm{EC}={\left(P\times h\right)}^{}/m$$
where EC is the energy consumption (kWh/t), P (voltage × electrical current) is the power of the equipment (kW), h is the time (h) spent during fibrillation, and m is the mass (t) of MFC/NFC processed in the fibrillator.

### 2.6. Microstructural Analysis of MFC/NFC

Following the methodology presented by Mascarenhas et al. [24], the MFC/NFC suspensions of each material were observed using an ultra-high-resolution (UHR) FEG scanning electron microscope Tescan-Clara (Kohoutovice, Czech Republic), under conditions of 10 KeV, 90 pA, with a distance of 10 mm work. Each sample was placed on double-sided carbon tape adhered to an aluminum sample holder (stubs). After an overnight period in a container with silica gel for drying, the samples were metalized with gold in a sputtering device (Balzers SCD 050). The micrographs were used to measure the MFC/NFC diameters. Results were obtained by the average of 200 measurements using the software Image J1.53e [25].

### 2.7. Suspension Stability

The MFC/NFC suspensions were diluted to 0.25% (*w*/*w*) in water, and 10 mL were transferred to test tubes. Images were obtained every hour during the first 8 h, ending the last capture after 24 h. Stabilities were measured at the total height of the liquid and suspension using the software Image J 1.53e [25], according to Equation (2), by the average of 5 measurements.
(2)$$St=\left(\frac{SP}{TH}\right)\times 100$$
where *St* is the suspension stability, *SP* is the height corresponding to the suspended particles, and *TH* is the total height of all the liquid in the tube.

### 2.8. Production of MFC/NFC films

The casting method was used for the production of MFC/NFC films, which consists of evaporating the solvent in acrylic Petri dishes with a diameter of 15 cm. Four films were prepared per raw material using 60 g of suspension at a concentration of 1% (*w*/*w*). The samples were dried for 5 days in an air-conditioned environment (20 ± 3 °C and 65%).

### 2.9. Physical Properties of the Films

The thickness of the films was measured using a digital calliper with a flat tip (0.001 mm), according to the standard Tappi T 411 om-15 [26]. The grammages were obtained according to the standard Tappi T 410 om-08 [27], weighing the samples on an analytical scale (0.001 g), and measuring their diameters with a digital calliper (0.001 mm). The apparent density was obtained by the ratio between the grammage and thickness of the films. The porosity was calculated according to Equation (3) [28]. Average values were obtained from 5 replicates.
(3)$$\Phi \;(\%)=1-{\left(\frac{\rho a}{1.54}\right)}^{\;}$$
where *ϕ* is the porosity, *ρa* is the apparent density (g cm^−3^), and 1.54 is the cellulose density (g cm^−3^).

### 2.10. Barrier to Water Vapor

The films’ water vapor transmission rate (WVTR) was performed by the gravimetric method, according to the standard ASTM e96-16 [29]. The samples with a diameter of 1.5 cm were sealed in a glass permeation cell containing 2/3 of its volume occupied by silica gel (0% relative humidity; without water vapor pressure) placed in a desiccator, which was kept at 25 °C and 100% relative humidity (Figure 1). The film was positioned on the lid of the glass bottle, forming a membrane between the outside and the inside of the permeability cell. The mass of the permeability cell was measured every 24 h for 8 consecutive days. The transmission rate values were calculated by Equation (4).
(4)$$\mathrm{WVTR}={\left(\frac{M}{t\times A}\right)}^{\;}$$
where *WVTR* is the water vapor transmission rate (g/m^2^ day), *M*/*t* is the angular coefficient of the graph obtained by linear regression of mass gain (g) versus conditioning time (days), and *A* is the exposed area of the sample (m^2^).

### 2.11. Test of Grease Resistance

The grease resistance of the films was carried out according to the standard Tappi T 559 cm-12 [30]. A drop of the solution was placed on the film’s surface, and the excess liquid was removed after 15 s with absorbent paper. The films were classified with scores from 1 to 12, according to the added solution. The solutions were ranked from 1, being “less aggressive,” less likely to pass through the film, and composed only of castor oil, to 12, being the “most aggressive,” composed of toluene (45% *v*/*v*) and n-heptane (55% *v*/*v*), with a greater tendency to pass through the film (Table 1).

### 2.12. Mechanical Properties of the Films

The tensile test was performed according to standard ASTM D882-12 [31] on a TATX2i Stable Micro Systems texturometer (Godalming, UK) equipped with a load cell with a capacity of 500 N. The initial distance between the grips was 50 mm, and the test was performed at a speed of 5 mm/s. Ten specimens of each treatment with 10 × 100 mm dimensions were used. Graphs with stress–strain curves were generated to observe the mechanical behavior of the films from the elastic phase to rupture.

### 2.13. Statistical Analyzes

Data were analyzed using descriptive statistics indicating the average and standard deviation. The results of the physical, barrier, and mechanical properties were submitted to analysis of variance (ANOVA). The Scott–Knott test was performed in case of significant differences between the averages of properties (*p* < 0.05).

## 3. Results and Discussion

### 3.1. Chemical Composition of the Fibers

The alkaline treatment and bleaching caused considerable changes in the levels of chemical components in the fibers (Table 2). There was a total removal of extractives from açaí fibers and partial removal from titica (~3.6%). For the cellulose content, a considerable increase occurred in the relative content of ~22% for açaí and ~19.6% for titica.

Commercial *Eucalyptus* sp. presented higher cellulose content and lower lignin and hemicellulose contents than açaí and titica fibers. Furthermore, total extractives in cellulosic pulps were not detected after alkaline treatment. As the pulps are commercial, the treatments were carried out in the company itself, so this raw material went through the traditional pulping and bleaching processes and laboratory alkaline treatment.

Regarding the amount of hemicelluloses, a 26% to 25% reduction was observed in açaí fibers. As for the titica vine, a decrease from 26.4% to 20% was marked. This is explained by the fact that hemicelluloses are easily hydrolyzed under alkaline conditions, which can hinder fiber fibrillation. This was reported by Dias et al. [32]; the authors found that when hemicellulose levels between 9 and 12% were obtained, a reduction in the coalescence of cellulose microfibrils could occur. Mascarenhas et al. [24] reported that hemicelluloses may facilitate cell wall swelling and may lead to reduced energy consumption during mechanical fibrillation.

Hemicelluloses can vary in structure according to the type of material in which they are embedded. Coutts and Warden [33] state that its percentage can increase after treatments depending on the specific type of fruit that originates the fibers and its growing conditions; reactions between fibers and the treatment solution cause such changes. In softwood, hemicelluloses mainly comprise galactoglucomannans [34]. In hardwood, the hemicellulose is primarily composed of a β-(1−4)-D-xylopyranose backbone with 4-O-methyl-α-D-glucuronic acids attached at C-2 of the xylose [35].

Despite the action of the treatments, the lignin was not wholly removed from the fiber structure. Total lignin was reduced from ~36% to ~17% for açaí fibers and from ~24.9% to ~15.4% for titica. Previous studies can also observe this same “median” removal behavior [6,7]. On the other hand, the presence of lignin in the structures can facilitate the deconstruction of the cell wall, making the fibrillation process more viable [1].

### 3.2. Energy Consumption

Titica vine fibrillation required an energy consumption of approximately ~25,000 kWh/t. The gel aspect was observed after 12 passages through the ultra refiner, at which time, an energy consumption of ~12,500 kWh/t was recorded. For açaí fibrillation, after 7 passages, there was a higher initial energy consumption compared to titica vine and Eucalyptus sp. However, the gel aspect was observed in the fifth pass, when the energy consumption was ~7700 kWh/t (Figure 2).

The commercial pulps of *Eucalyptus* sp. required lower energy consumption (~4100 kWh/t), with gel formation in the fourth pass. This energy consumption to produce MFC/NFC is a critical factor in enabling the competitive commercialization of the industrial production of these materials and their derivatives so that they can replace petroleum-based polymers [1]. According to Desmaisons et al. (2017), since 2008, pre-treatments have reduced energy consumption in the fibrillation process from 30,000 kWh/t to less than 2000 kWh/t.

The gradual delamination of the cell wall may be a factor that explains the higher energy consumption in titica vine fibrillation. This characteristic stems from the lower water retention capacity of the suspension to the detriment of the lower hemicellulose content. This contributes to obtaining a lower viscosity for the MFC/NFC suspensions, which justifies the need to employ more shear cycles in the ultra refiner.

The increase in viscosity throughout the fibrillation process results from the disintegration of the fibril bundles, showing a more significant formation of a three-dimensional NFC network with a more consistent gel structure [36]. In general, the fibrillation of the raw material depends on the resistance offered by its cell wall. This resistance is directly related to the basic density of the titica fibers (0.54 g/cm^3^), which is higher in relation to açaí (0.32 g/cm^3^) [9,37]. In other words, the thickness of the cell walls of titica vine is greater than those observed for açaí; therefore, in the material with greater density, there will be more bundles of microfibrils that will offer greater resistance to rupture by the discs of the ultra refiner.

### 3.3. Microstructural Analysis of MFC/NFC

The highest amount of MFC/NFC corresponds to diameter classes from 30 to 45 nm (35%) for *Eucalyptus* sp. As for açaí, the highest frequency was observed for the range between 45 and 60 nm (30%), and for titica liana, the highest frequency was obtained for the diametric class of 75 to 90 nm (29%) (Figure 3).

In the micrographs, it was observed the significant agglomerations of MFC/NFC, mainly for açaí and titica vine (Figure 4). The high aspect ratio and the establishment of solid hydrogen bonds after mechanical fibrillation can explain this.

These results indicate that the commercial pulp of *Eucalyptus* sp. presented better fibrillation and individualization of the fibrils. According to Scatolino et al. [9], smaller diameters of MFC/NFC allow for the greater interweaving of structures due to the greater surface area, that is, a more significant number of NFC, providing films with higher density and better physical, mechanical, and barrier properties.

Mascarenhas et al. [24] also observed this behavior when evaluating the mechanical fibrillation process in *Eucalyptus,* and *Pinus* treated with sodium silicate. The authors indicated that because the fibers of *Pinus* have a higher aspect ratio, there is a tendency for the MFC/NFC network to present itself in complex tangles and with greater length, which contributed to greater energy consumption during mechanical fibrillation.

It can be said that the MFC/NFC from titica, due to the larger diameters, presented a smaller surface area. A higher level of aggregation can limit the interaction with polymer matrices through hydrogen bonds, reducing the tenacity structure of bionanocomposites [8]. Furthermore, larger dimensions of MFC/NFC may imply greater mass, causing the increased sedimentation of suspensions [24].

### 3.4. Suspension Stability

When analyzing the sedimentation, it was possible to notice a decreasing trend in the stability of the suspension over the initial 8 h and after 24 h (Figure 5). The açaí MFC/NFC suspensions showed less stability compared to the titica vine and Eucalyptus sp. commercial, reaching final stability close to ~70% after 24 h. As *Eucalyptus* sp. only have traces of lignin and higher hemicellulose contents, the hydroxyl groups of carbons 2, 3, and 6 of the amorphous regions of the cellulosic chain are more exposed and confer greater electrostatic repulsion, keeping the particles in suspension for longer [24].

Thus, the açaí MFC/NFC suspension sedimentation may have occurred due to a considerable amount of lignin in its structure (17%) and hemicelluloses (25%) in the raw material. Non-cellulosic components can vary in type, polarity, cohesion energy, and molecular weight according to the kind of raw material in which they are inserted [38].

The study by Dias et al. [1] found stability of 53% for NFC of bleached mandacaru fibers (*Cereus jamacaru*) after 24 h of evaluation, which indicates that the type of lignocellulosic material to be fibrillated is a factor that can cause variations in the characteristics of the produced NFC. When the suspension is significantly diluted, the particles that make up the MFC/NFC will have a random arrangement due to Brownian motion that will keep them dispersed, thus increasing their stability [39].

When observing the sedimentation of MFC/NFC from titica, it is considered that the suspensions remained stable due to the difficulty in identifying the decanted and suspended parts. The calculated stability value for this raw material was around ~99%. However, a visible amount of suspension settled to the bottom of the test tubes (red square), reaching a final proportion of approximately 9% after 24 h. This fact can be explained by the large proportion of MFC about NFC in suspensions.

The MFC/NFC of the titica vine suspensions showed larger diameters of the structures (see Figure 3 and Figure 4). This may have contributed to the increase in aggregation and the increase in the mass of the tangles. In addition, the amount of lignin present in this material may have contributed to the rise in the overall molar mass of the material. Notably, suspensions with high stability may present greater ease of redispersion, mainly if the suspension contains lignin levels in the composition. Kim et al. [40] reported that NFC having specific levels of lignin showed greater ease of redispersion and maintenance of their properties after drying the material. This is an attractive characteristic that can be attributed to the MFC/NFC of açaí and titica, considering that the drying of this material can facilitate its transport and commercialization due to the reduction in logistical costs.

### 3.5. Physical Properties of the Films

Açaí MFC/NFC films were thicker than titica and *Eucalyptus* sp. (Table 3). The lowest grammage was observed for titica vine films (28.2 g/m^2^). The films of *Eucalyptus* sp. showed higher apparent density and, consequently, lower porosity, indicating that the mechanical fibrillation process efficiently reduced fiber dimensions, resulting in a uniform and more compact films. These factors strongly influence other properties, such as mechanical and permeability.

As previously seen, the filmogenic suspensions of *Eucalyptus* sp. have a higher frequency of structures with a nanoscale diameter (see Figure 3). This feature confers greater surface area and contributes to the formation of more significant amounts of electrostatic interactions and connections between MFC/NFC more efficiently. Butchosa and Zhou [41] reported that the increase in hydrogen bonds in the NFC network decreases empty spaces, increases the apparent density of the films, and, consequently, guarantees superior barrier and mechanical properties to the films produced.

The observed variations in thickness and grammage can be explained, in addition to the type of material used, by aspects inherent to the film production method. Although the casting method has been widely used in research with different lignocellulosic materials for the production of MFC/NFC films, there are limitations, such as difficulty in producing films in larger dimensions, long drying time, thickness heterogeneity, and wrinkling due to the non-application of a vacuum [42].

The MFC/NFC films produced with titica showed a higher water vapor transmission rate (WVRT), while the MFC/NFC films made from açaí and commercial pulp did not differ (Figure 6). The apparent density and porosity of the MFC/NFC films of açaí and titica did not vary. However, among the raw materials, titica obtained the lowest grammage, as previously mentioned (see Table 3).

Density and grammage can be highlighted when analyzing the barrier mechanisms, as they are strongly correlated. The ideal structure of MFC/NFC networks is a compact and complex shape, presenting an obstacle to water vapor diffusion. As already noted, the large amount of micro-scale fibrils may have affected the three-dimensional structure of the titica vine network, reducing the number of interfibrillar interactions and allowing for the formation of isolated aggregates.

The existence of more significant proportions of empty spaces in the microstructures of the films can explain the high values of WVTR. This can be observed in the high porosity value, as the appearance of regions with empty spaces may have facilitated water vapor passage [43].

Even so, the WVTR values of the MFC/NFC films of açaí and Eucalyptus sp. were close to those found in the literature for films produced with other lignocellulosic materials [24]. In the study by Guimarães et al. [6], when evaluating films produced with NFC from banana pseudostem residues, found WVTR values of 519 and 497 g day^−1^ m^−2^ after 20 and 40 passages, respectively, through the ultra refiner.

Likewise, Stark [44] found WVTR results close to the value obtained for titica films. When evaluating CNF films from bleached hardwoods and softwoods, this author found values of 606 and 686 g day^−1^ m^−2^, respectively. This indicates the potential of titica vine films for packaging applications, with advances in pre-screening renewable lignocellulosic materials to replace petroleum-based polymers in multilayer packaging.

The primary function of packaging for plant-based fresh foods is to prevent or reduce this moisture transfer and the surrounding atmosphere. Therefore, the lower the film’s water vapor permeability, the better it will be to optimize and make the product viable for commercialization [8].

### 3.6. Test of Grease Resistance

Grease resistance was measured according to the test based on 12 different oil solutions of different concentrations (Figure 7). The films showed high resistance to grease due to the “more aggressive” solution (score 12), that is, the one with a high penetration capacity in the interfibrillar structure of the film, composed of toluene and n-heptane, not crossing the samples.

In addition to offering information on resistance to the penetration of oily substances, the grease resistance test also provides evidence to infer whether the package in question could receive ink coatings in the case of surface printing or even evaluate adhesive applications, which usually look slimy and sticky. This characteristic could culminate in a strong point since it would eliminate the need to apply seals with information about the product; that is, nutritional information, bar codes, or composition of ingredients could be written on the packaging itself.

This would imply the use of less material per package and also a reduction in final costs. Tayeb et al. [45] found high grease resistance when analyzing lignin-containing cellulose nanofibril packaging, indicating that all MFC/NFC films in the present study have the potential to be used as packaging materials, generating an efficient oil barrier. Packaging that has a high barrier to fat penetration and the ability to retain viscous oils may be ideal for wrapping some specific foods, such as grains and bakery products [24].

The resistance offered by the surface of the film to the spreading of water is also an important parameter to be evaluated in the characterization of the films when sent for packaging purposes. The average values of the contact angle of the water droplet with the surface were higher for the MFC/NFC films of *Eucalyptus* sp. concerning the angles formed on the açaí and titica films (Figure 8).

The MFC/NFC films of açaí and titica showed hydrophilic behavior without differences between them, indicating a greater susceptibility to wetness (<90°). In the case of the two raw materials mentioned, lignin may have affected the connectivity between the fibrils, providing water penetration and greater spreading of liquids on the surface of the films. Hydrophilic films can also be helpful when the objective is to obtain characters that paint and coloring pigments can spread.

The contact angle values of *Eucalyptus* sp. were classified as hydrophobic and partially lipophilic material since the contact angle formed was 95.2°, which is within the range of 90–180° [46,47]. A plausible explanation may be related to the organization of the nanostructure of *Eucalyptus* sp. films, which is more compact, denser, and less porous. These characteristics make it difficult for water to penetrate the internal structures.

These results are essential for possible applications of these materials in uses that may require reduced contact angle values (the application of adhesives, application of paints, and adsorption of liquids) [6]. On the other hand, using films with a hydrophilic nature for the production of packaging can represent difficulties in storing certain types of products, mainly perishable foods [48].

### 3.7. Mechanical Properties of the Films

Films produced with MFC/NFC from *Eucalyptus* sp. showed greater tensile strength than the other raw materials (Figure 9). The stress × strain curves highlighted the superior mechanical strength of *Eucalyptus* sp. compared to the others studied (Figure 9b).

These results demonstrate that films produced from commercial pulp MFC/NFC have a more homogeneous structure with few empty spaces. This observation is corroborated by the high apparent density (1.16 g/cm^3^) and low porosity (24.7%). The more upright the slope of the stress–strain curve, the greater the tensile stiffness of the films [49].

The açaí and titica species used in this research were obtained in the raw state and treated in laboratory processes since the extraction of the vegetal. The açaí fibers removed from the seed underwent grinding, alkaline treatment, and bleaching, unlike the titica vine, which underwent only grinding, followed by alkaline treatment without bleaching. The absence of the bleaching process may have affected the final quality of titica films since bleaching can leave lignin and hemicellulose contents at ideal values for fibrillation. Hemicellulose contents between levels of 9 to 12% can be favorable to fibrillation since these polysaccharides prevent the coalescence of cellulose microfibrils [32].

Some lignin content is reported to increase film toughness, tensile index, and Young’s modulus [50]. Contrarily, high lignin content means fewer fibrillated fibers, which can adversely affect mechanical properties [3]. Lignin works as an excellent adhesive within the lignocellulosic structure, but between interfibrillar structures, after the rupture of the cell wall, it can cause poor intertwining, impairing the tensile properties.

Rojo et al. [51] found a decrease in fiber diameters with residual lignin after the fibrillation and softening of the lignin. Due to its amorphous nature, the lignin allowed for a bonding effect, filling the empty spaces between the NFC and providing smoother surfaces for the nanostructured papers. The result obtained from açaí films was superior to that from titica. Despite the apparent density and porosity between the films obtained from these two raw materials, there were no differences. Still, the grammage was higher for the açaí films, which may explain the greater resistance. Furthermore, the tensile strength of the açaí films was superior to the results found in the literature for *Cereus jamacaru* fiber films (51.0 MPa) [1]. Guimaraes et al. [6] produced NFC films from banana pseudostems after 20 and 40 passes through the Supermasscolloider grinder and showed tensile strengths of 46 and 51 MPa, respectively. These results were close to the values in this study for titica films.

## 4. Conclusions

In this study, MFC/NFC films were produced with forest residues from the Amazonia and evaluated to possibly replace plastic packaging. The açaí and titica vine fibers significantly resisted removing non-cellulosic components from the fiber. *Eucalyptus* sp pulps showed higher fibrillation and MFC/NFC individualization. As for the titica vine, mechanical fibrillation was not as effective, impairing some properties of the films, which showed a higher WVTR (637.3 g day^−1^ m^−2^) and lower tensile strength (46.2 MPa). All films showed strong resistance to grease. The films produced in the study showed potential for application in packaging for light and low-humidity products due to their physical, mechanical, and barrier properties. New studies are needed to investigate pre-treatments or alternative and ecologically correct fibrillation methods that are viable for reducing energy consumption. This can contribute to enabling the application of these materials more efficiently and improve the fibers’ mechanical fibrillation process, especially for açaí fibers.

## Figures and Tables

**Figure 1 polymers-15-03646-f001:**
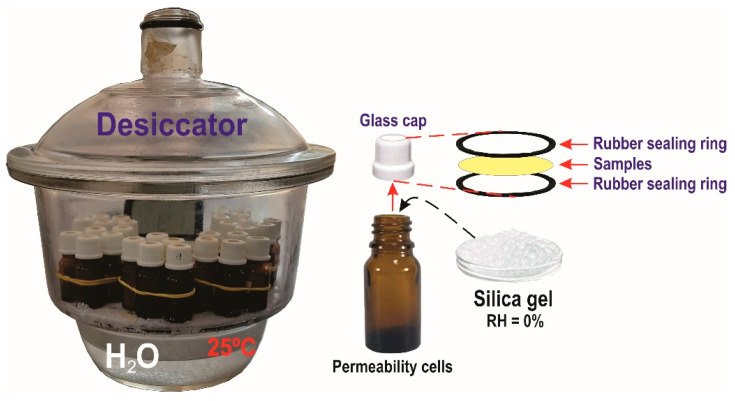
Test for measuring the water vapor barrier of the films.

**Figure 2 polymers-15-03646-f002:**
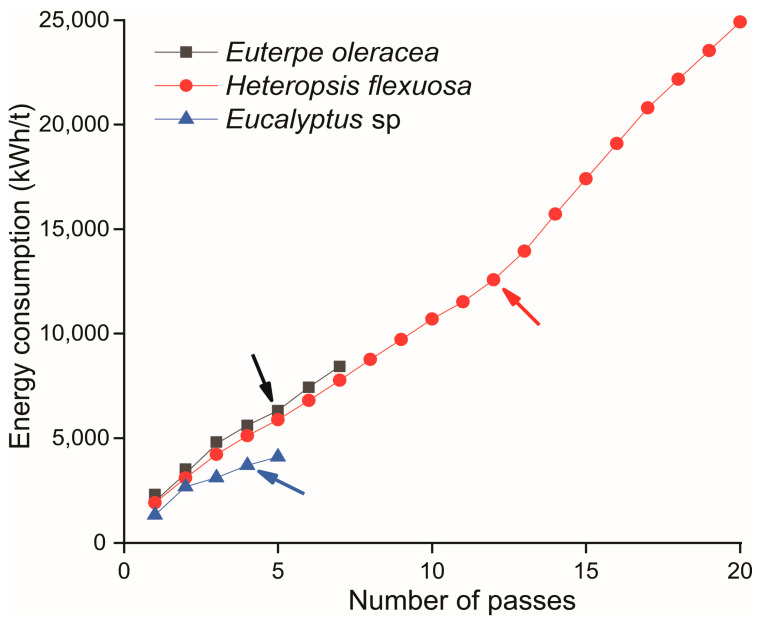
Energy consumption with increasing mechanical fibrillation cycles of each material: *Euterpe oleracea* (açaí); *Heteropsis flexuosa* (titica); and *Eucalyptus* sp. The arrows indicate the passage where the MFC/NFC started to gel aspect.

**Figure 3 polymers-15-03646-f003:**
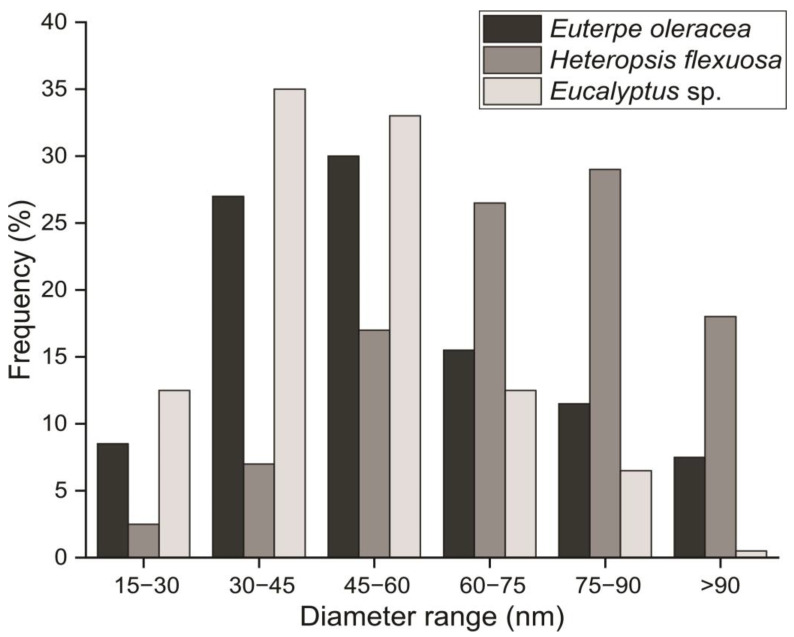
MFC/NFC diameter distribution of the studied raw materials: *Euterpe oleracea* (açaí); *Heteropsis flexuosa* (titica); and *Eucalyptus* sp.

**Figure 4 polymers-15-03646-f004:**
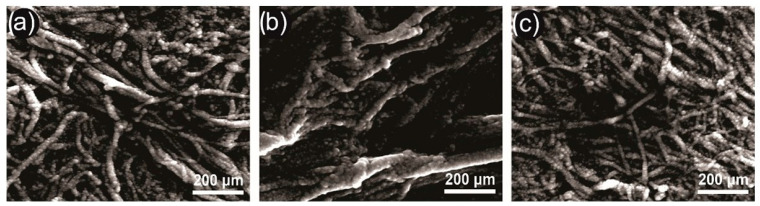
Typical SEM images of the MFC/NFC from: (**a**) *Euterpe oleracea* (açaí); (**b**) *Heteropsis flexuosa* (titica); and (**c**) *Eucalyptus* sp.

**Figure 5 polymers-15-03646-f005:**
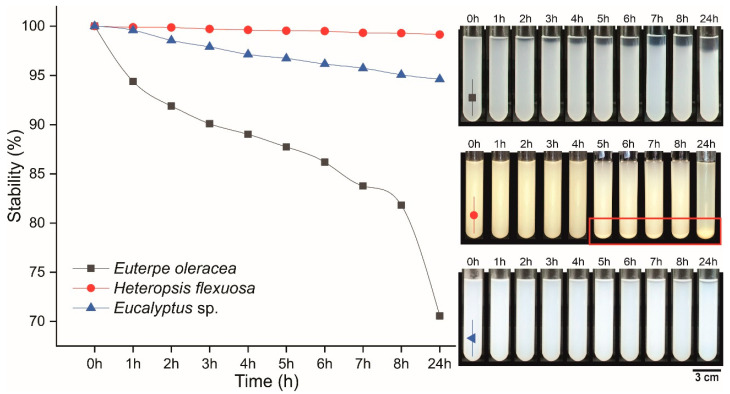
Stability of suspensions produced from different raw materials: *Euterpe oleracea* (açaí); *Heteropsis flexuosa* (titica); and *Eucalyptus* sp.

**Figure 6 polymers-15-03646-f006:**
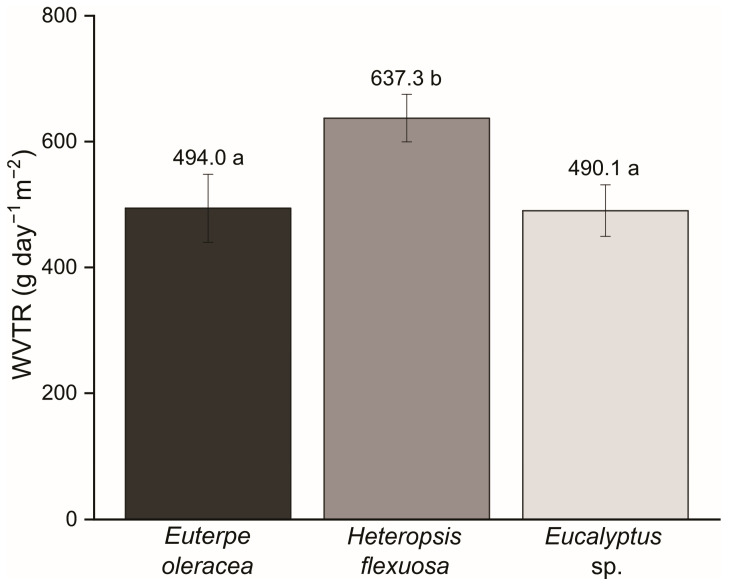
Water vapor transmission rate (WVTR) of the films produced from MFC/NFC obtained with different raw materials: *Euterpe oleracea* (açaí); *Heteropsis flexuosa* (titica); and *Eucalyptus* sp. Averages followed by the same letter do not differ by the Scott–Knott test (ρ < 0.05).

**Figure 7 polymers-15-03646-f007:**
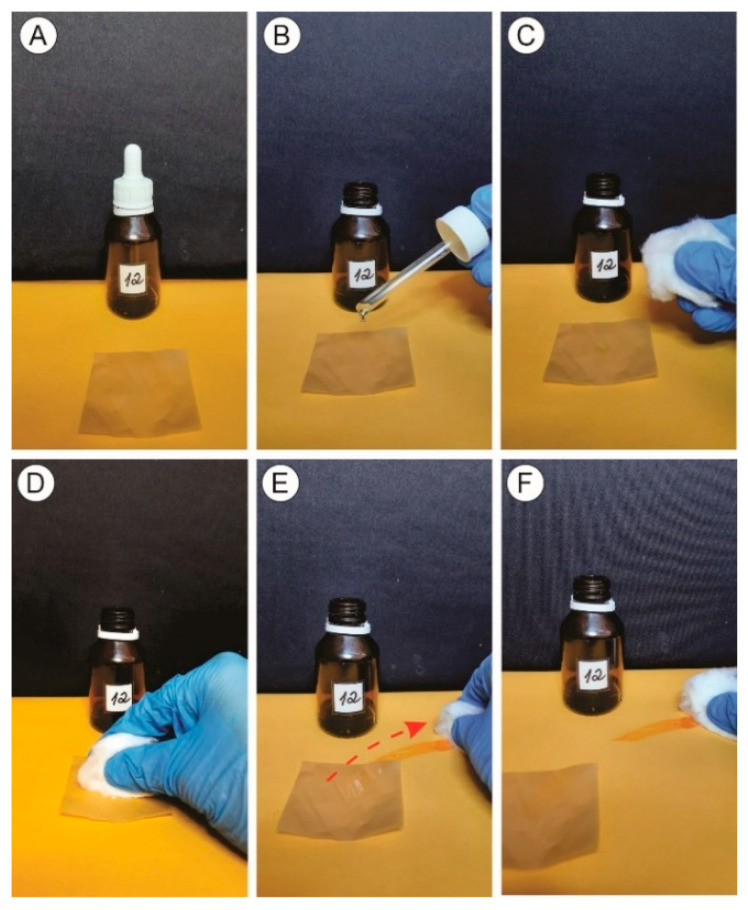
Illustrative of the grease resistance test. (**A**) Sample with the “most aggressive” solution (score 12). (**B**) Drop of the solution is placed on the surface. (**C**) Kept in contact for 15 s. (**D**) A piece of cotton is passed over the region where the drop acts. (**E**) A movement is extended until the “solution trail” (red arrow) reaches the yellow paper positioned below the sample. (**F**) The yellow paper below is analyzed for traces of solution that possibly passed through the sample.

**Figure 8 polymers-15-03646-f008:**
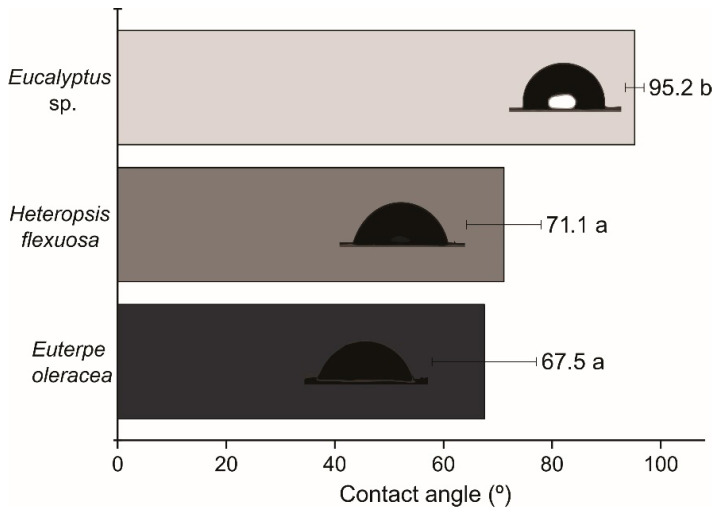
Contact angle of the water drop with the surface of films produced with MFC/NFC obtained with different raw materials: *Euterpe oleracea* (açaí); *Heteropsis flexuosa* (titica); and *Eucalyptus* sp. Averages followed by the same letter do not differ by the Scott-Knott test (ρ < 0.05).

**Figure 9 polymers-15-03646-f009:**
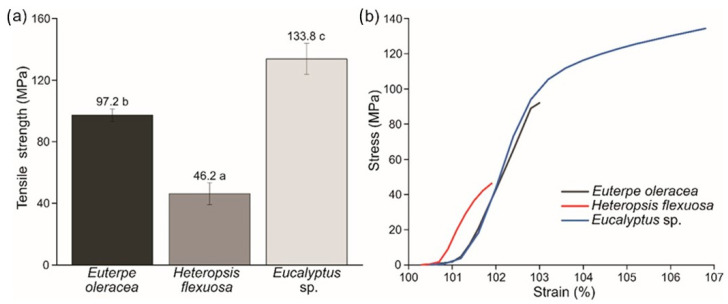
(**a**) Tensile strength of the films. (**b**) Stress × strain curves of the films produced with MFC/NFC obtained with different raw materials: *Euterpe oleracea* (açaí); *Heteropsis flexuosa* (titica); and *Eucalyptus* sp. Averages followed by the same letter do not differ by the Scott-Knott test (ρ < 0.05).

**Table 1 polymers-15-03646-t001:** Score and composition of the solutions for the test of grease resistance.

Score	Castor Oil	Toluene	n-Heptane
	---------------------------%-------------------------		
1	100	0	0
2	90	5	5
3	80	10	10
4	70	15	15
5	60	20	20
6	50	25	25
7	40	30	30
8	30	35	35
9	20	40	40
10	10	45	45
11	0	50	50
12	0	45	55

**Table 2 polymers-15-03646-t002:** Chemical constitution of *Euterpe oleracea* (açaí), *Heteropsis flexuosa* (titica), and commercial *Eucalyptus* pulp.

Component	*Euterpe oleracea* (açaí)		*Heteropsis flexuosa* (titica)		* Commercial *Eucalyptus* Pulp
	*In Natura*	Bleached	*In Natura*	Alkali-Treated	
	-----------------------------(%)-----------------------------				
Total extractives	4.0 ± 0.2	-	5.9 ± 0.4	2.3 ± 0.1	-
Insoluble lignin	36.0 ± 0.6	17.0 ± 3.6	24.9 ± 1.7	15.4 ± 1.1	0.14 ± 1.1
Holocellulose	60.0 ± 2.0	81.0 ± 2.8	68.4 ± 1.4	80.6 ± 1.1	90.0 ± 1.1
Cellulose	34.0 ± 1.8	56.0 ± 0.2	42.0 ± 1.2	60.6 ± 2.2	76.0 ± 2.2
Hemicelluloses	26.0 ± 1.9	25.0 ± 0.1	26.4 ± 2.5	20.0 ± 1.8	14.0 ± 1.8
Ash	1.6 ± 0.2	0.3 ± 0.1	1.3 ± 0.1	1.6 ± 0.05	0.03 ± 0.01

* Commercial pulps already obtained in the bleached form.

**Table 3 polymers-15-03646-t003:** Average values and standard deviation of thickness, grammage, apparent density, and porosity of films obtained from different raw materials: *Euterpe oleracea* (açaí); *Heteropsis flexuosa* (titica); and *Eucalyptus* sp.

Raw Material	Thickness (µm)	Grammage (g/m^2^)	Apparent Density (g/cm^3^)	Porosity(%)
*Euterpe oleracea*	43.0 ± 9.0 * b	33.7 ± 1.9 b	0.82 ± 0.15 a	46.7 ± 3.0 b
*Heteropsis flexuosa*	32.4 ± 1.7 a	28.2 ± 2.5 a	0.79 ± 0.05 a	48.2 ± 3.4 b
*Eucalyptus* sp.	31.0 ± 1.0 a	35.5 ± 2.3 b	1.16 ± 0.03 b	24.7 ± 4.7 a

Averages followed by the same letter do not differ by the Scott–Knott test (ρ < 0.05). * Standard deviation.

## Data Availability

All relevant information has been shown in the article.

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
