# Peer review of "Developing a Biodegradable Film for Packaging with Lignocellulosic Materials from the Amazonian Biodiversity"

_polymers, 2023, doi:10.3390/polym15173646_

Round 1

Reviewer 1 Report

The paper describes formation of biofilms using two different Amazonian plants and the results were compared with the commercial preparation of Eucalyptus. The authors have also presented extensive characterization of the films. Some aspects need to be addressed.

1.     Language quality is not acceptable. Authors need to revise the paper thoroughly.

2.     Title is confusing: Modify it.

3.     L24-26: Write rationale of this study with respect to the advantage of biodegradable films using renewable sources.

4.     L28-30: State why these three plants were selected.

5.     In abstract, present all the key findings.

6.     L48: Write about some utility/significance of MFC/NFC

7.     L62-69: The section needs revision as the authors also used hydrogen peroxide for bleaching purpose.

8.     L132: Correct the formula

9.     Table 2: Correct the spellings. Also, state in caption, why Native Eucalyptus composition is not given.

10.  L312: Correct to remove formatting issue.

There are several grammatical errors and spelling mistakes.

Author Response

Responses to reviewer comments are attached.

Reviewer 2 Report

Authors - Danillo Wisky Silva, Felipe Gomes Batista, Mário Vanoli Scatolino, Adriano Reis Prazeres Mascarenhas, Dayane Targino de Medeiros, Gustavo Henrique Denzin Tonoli , Daniel Alberto Álvarez La , Francisco de Tarso Ribeiro Caselli, Tiago Marcolino de Souza and Francisco Tarcísio Alves Junior

Article “Production of biodegradable films for nanostructured packaging with lignocellulosic materials from the Amazonian biodiversity”

The authors presented a good review paper on the current state of research on cellulose materials obtained from various plants. However, industrial cellulose films for food products have been produced for a long time, the so-called cellophane. In this regard, the novelty of the authors' work is unclear. What result do they want to achieve, what new characteristics? Or can the fibrillation method be applied for the first time for these materials?

Please answer the following questions:

1.    It is necessary to describe the process of fibrillation in more detail. Why was this particular method used to produce fibers?

2.    In the Materials and Methods section, when calculating Suspension stability, it is not indicated with which liquids the suspensions were diluted. It should be specified. How did the authors evaluate the solubility of their materials? How mono- or polydisperse are the samples? Could it be that the low-molecular fraction was in solution? How correct is this method of estimation of suspension stability and what is its error?

3.    Many of the authors' statements, in particular concerning the formation of hydrogen bonds, must be confirmed experimentally, for example, by the IR method. It would also be useful to determine the particle sizes in the suspension at different concentrations.

4.    In the Materials and methods section, the oils used are not indicated in the Test of Grease Resistance section.

Author Response

(The authors gave the same response as above.)

Round 2

Reviewer 2 Report

The authors answered the questions and comments in detail. The necessary changes and additions have been made to the article. The article can be published